# The Mechanisms of Altered Blood–Brain Barrier Permeability in CD19 CAR T–Cell Recipients

**DOI:** 10.3390/ijms25010644

**Published:** 2024-01-04

**Authors:** Soniya N. Pinto, Giedre Krenciute

**Affiliations:** 1Department of Diagnostic Imaging, St. Jude Children’s Research Hospital, Memphis, TN 38105, USA; 2Department of Bone Marrow Transplantation & Cellular Therapy, St. Jude Children’s Research Hospital, Memphis, TN 38105, USA; giedre.krenciute@stjude.org

**Keywords:** blood–brain barrier, CD19 CAR T cells, neurotoxicity

## Abstract

Cluster of differentiation 19 (CD19) chimeric antigen receptor (CAR) T cells are a highly effective immunotherapy for relapsed and refractory B-cell malignancies, but their utility can be limited by the development of immune effector cell-associated neurotoxicity syndrome (ICANS). The recent discovery of CD19 expression on the pericytes in the blood–brain barrier (BBB) suggests an important off-target mechanism for ICANS development. In addition, the release of systemic cytokines stimulated by the engagement of CD19 with the CAR T cells can cause endothelial activation and decreased expression of tight junction molecules, further damaging the integrity of the BBB. Once within the brain microenvironment, cytokines trigger a cytokine-specific cascade of neuroinflammatory responses, which manifest clinically as a spectrum of neurological changes. Brain imaging is frequently negative or nonspecific, and treatment involves close neurologic monitoring, supportive care, interleukin antagonists, and steroids. The goal of this review is to inform readers about the normal development and microstructure of the BBB, its unique susceptibility to CD19 CAR T cells, the role of individual cytokines on specific elements of the brain’s microstructural environment, and the clinical and imaging manifestations of ICANS. Our review will link cellular pathophysiology with the clinical and radiological manifestations of a complex clinical entity.

## 1. Introduction

Cluster of differentiation 19 (CD19) is a surface glycoprotein expressed on both physiologic and neoplastic B cells and represents a valuable target for immunotherapies against B-cell malignancies [1]. CD19-directed chimeric antigen receptor (CAR) T cells are a promising therapeutic strategy for both adult and pediatric patients with relapsed or refractory B-cell malignancies. These include patients with B-cell acute lymphoblastic leukemia, chronic lymphocytic leukemia, non-Hodgkin’s lymphoma, and multiple myeloma, who have not responded or relapsed following conventional chemotherapy and/or a hematopoietic stem cell transplant. The T cells are autologously harvested from the patient, stimulated with a cytokine cocktail containing CD8 and CD28, and treated with viral vector agents to express CARs directed against CD19 on their cell surface. The CAR consists of an extracellular domain with a single-chain variable fragment derived from an anti-CD19 antibody, a transmembrane domain, and an intracellular co-stimulatory domain derived from either 4-1BB or CD28 linked to CD3ζ. Once CAR T cells are intravenously infused back into the patient, their engagement with CD19 activates the intracellular co-stimulatory and CD3ζ domains and the synthesis of proteins such as perforin and granzyme. Perforin increases the porosity of neoplastic B cells and granzyme influxes into neoplastic B cells to activate intracellular caspases, thereby initiating apoptosis [2].

Although highly effective in targeting neoplastic B cells, CD19 expression on off-target cells can cause a range of complications. CD19 is also expressed on physiologic B cells, thus their depletion results in hypogammaglobulinemia and increased susceptibility to opportunistic infections [3]. Recent single-cell RNA sequencing (scRNA-seq) analyses identified CD19 on pericytes in the human blood–brain barrier (BBB); pericytes play a critical role in maintaining the integrity of the BBB [4]. This recent discovery has provided important insight into the mechanisms of immune effector cell-associated neurotoxicity syndrome (ICANS), which has been reported with higher frequency and severity in recipients of CD19 CAR T cells, as compared to recipients of CAR T cells with different targets (e.g., CD22) [5]. In addition, CD19 CAR T–cell therapy activates the monocyte/macrophage system and the systemic release of inflammatory cytokines, such as granulocyte monocyte colony-stimulating factor (GM-CSF); interleukins (IL)-1β, IL6, and IL15; and interferon-gamma (IFN-γ), which can further increase the permeability of the BBB [6,7]. Once trafficked across the BBB, these cytokines activate a host of neuroinflammatory responses in the brain microenvironment, which manifests clinically as ICANS [8].

The goal of this narrative review is to systematically consider the microstructure of the BBB, while highlighting the role of each element in the pathogenesis of ICANS and detailing the mechanism of cytokine release after CD19 CAR T–cell infusion, the impact of individual cytokines on the BBB, and the cascade of neuroinflammatory responses that follow. We will conclude our review by linking these cellular mechanisms to the complex and varied clinical and radiological features of ICANS.

## 2. The Blood–Brain Barrier

The BBB is a complex neurovascular structure composed of an inner layer of highly specialized endothelial cells, surrounded by pericytes embedded in the endothelial basement membrane and an outer layer of astrocyte foot processes [9]. The inner endothelial cells are the first to develop, with immature angioblasts infiltrating the central nervous system (CNS) on approximately embryonic day (E) 12. This is followed by the arrival of pericytes, which stimulate the expression of tight junction strands between the individual endothelial clefts [10]. These tight junction strands anchor to the cytoskeleton of the endothelial cells via tethering to intracellular actin filaments [11]. The tight junction strands consist of three major types of proteins: claudins, MARVEL/D3, and immunoglobulin-like junctional adhesion molecules. Although claudins and MARVEL/D3 proteins play a complementary role in maintaining the transportation of solutes across the cell membrane, the junctional adhesion molecules play a key role in immune cell trafficking across the BBB [11]. Junctional adhesion molecules act in conjunction with basolateral vascular endothelial cadherins and cellular adhesion molecules to facilitate the transport of leukocytes across the BBB. The pro-inflammatory cytokines released during CD19 CAR T–cell therapy, particularly IL-1β, indirectly decrease the expression of these tight junction proteins, thereby disrupting the integrity of the BBB [12].

Immediately external to the vascular endothelium are the pericytes, which play a key role in vessel wall regeneration [13]. Pericytes preferentially contact the endothelial cells at tight junctions, while also positioning themselves at the luminal interfaces of the astrocyte foot processes [14]. A study by Parker et al., employing scRNA-seq analyses of more than 2000 samples of human prefrontal cortex cells, identified CD19 and CD81, the latter of which is essential to cell surface expression of CD19 on human pericytes. CD19 is expressed early in brain development, concurrently with the development of pericytes, and persists throughout adulthood. By comparison, scRNA-seq analysis of mouse mural models demonstrated relatively lower levels of CD19 expression, which most likely accounts for the lower incidence of ICANS in mouse models [4].

The outermost layer of the BBB consists of astrocyte foot processes that ensheath more than 90% of the capillary surface area in the brain. The predominant role of astrocyte foot processes is to maintain the BBB via the wingless-related integration site (Wnt) and Sonic Hedgehog (SHH) pathways [15]. The expression of the Wnt ligand enhances that of glial-derived neurotropic factor, which is critical for maintaining tight junction integrity [16]. In contrast, the SHH ligand binds to cell surface G-protein–coupled receptors and downregulates the expression of pro-inflammatory IL-8 and monocyte chemoattractant protein-1, while upregulating tight junction protein expression [17]. Although the exact mechanism of astrocytic injury in ICANS is yet to be determined, the elevated pro-inflammatory cytokines may disrupt the protective mechanisms of the Wnt- and SHH-signaling pathways, leading to osmotic dysregulation and neurotoxicity [18].

## 3. The Role of Cytokines in the Breakdown of the Blood–Brain Barrier

Once transfused into the recipient, CAR T cells bind with high affinity to CD19 on the surface of neoplastic B cells. This engagement activates the monocyte/macrophage system, which results in the systemic release of several cytokines, including GM-CSF, IFN-γ, IL-1β, IL-6, IL-10, IL-15, and angiopoietin-2 [19]. The mechanisms of each of these cytokines in the BBB breakdown and subsequent neuroinflammation are briefly summarized in Figure 1 and discussed in greater detail below.

### 3.1. Granulocyte-Monocyte Colony-Stimulating Factor

The engagement of CD19 with CAR T cells stimulates the release of GM-CSF from the monocyte/macrophage system, which increases both BBB permeability and subsequent neuroinflammation. Studies in preclinical models of neuroinflammation have noted an increased migration of GM-CSF–stimulated monocytes across the BBB, as compared to nonstimulated control monocytes; this increased migration has been linked to a significantly higher secretion of tumor necrosis factor-alpha (TNF-α) by the stimulated monocyte population [20,21]. TNF-α, in turn, increases the expression of leukocyte adhesion molecules, such as intercellular adhesion molecule 1 (ICAM-1) and vascular cell adhesion molecule 1 (VCAM-1), which enable the attachment and migration of stimulated monocytes across the BBB [22]. Once trafficked across the BBB, the GM-CSF–stimulated monocytes mature into macrophages and secrete reactive oxygen species that damage the cytoskeleton of neurons and glial cells [23]. In a preclinical study by Sterner et al., a mouse model implanted with leukemic blasts and treated with human CD19 CAR T cells developed hunched posture and motor weakness, with elevated serum levels of GM-CSF and increased contrast enhancement on brain magnetic resonance images (MRIs), compatible with neurotoxicity. The subsequent administration of lenzilumab, a GM-CSF–neutralizing antibody, decreased the contrast enhancement on brain MRIs and reduced the infiltration of activated macrophages into the brain microenvironment, while preserving the antitumor activity of the CD19 CAR T cells [24]. Clinical studies in both adult and pediatric CD19 CAR T–cell recipients have mirrored these results, with elevated serum levels of GM-CSF noted during the acute neurotoxicity phase [25,26,27]. To date, however, only one clinical study, by Gust et al. in 2019, involving 43 pediatric and young adult recipients of CD19 CAR T–cell therapy, has investigated GM-CSF levels in cerebrospinal fluid during the acute neurotoxicity phase and demonstrated no significant elevation [28]. Early evidence suggests the primary role of GM-CSF as an enabler of activated monocyte/macrophage transport across the BBB, with the subsequent neuroinflammation resulting from the local release of reactive oxygen species. Furthermore, larger-scale clinical studies investigating cerebrospinal fluid levels of GM-CSF are needed to validate this hypothesis. In addition, the potential value of lenzilumab in counteracting neuroinflammation while preserving CAR T–cell function is currently under investigation in a clinical trial (NCT04314843).

### 3.2. Interferon-γ

IFN-γ is a key cytokine secreted by actively proliferating CD19 CAR T cells. Preclinical models of neuroinflammation have shown that IFN-γ induces gene expression within the endothelial cells of the BBB and increases the expression of ICAM-1 and VCAM-1, which most likely help traffic CD19 CAR T cells into the CNS [29]. Studies of the effects of IFN-γ, once trafficked across the BBB, have yielded conflicting results in preclinical models, depending on the disease process being studied. In models of autoimmune encephalitis, IFN-γ stabilizes the endothelium and decreases leukocyte infiltration; however, in models of reovirus infection, IFN-γ increases permeability [30,31]. Preclinical models of ICANS in which the neuroinflammatory effect of IFN-γ has been evaluated are currently lacking, but indirect evidence suggests that the cytokine has a pro-inflammatory role. An investigation by Zhang et al. in 2022 reported the effects of an IL-6/IFN-γ double-knockout CD19 CAR T–cell product in peripheral monocyte blood cultures; they found that their product induced significantly lower levels of pro-inflammatory cytokines than a standard CD19 CAR T–cell product did, while retaining antitumor activity [32]. More preclinical studies on the effects of IFN-γ on specific elements of the brain microenvironment in ICANS are needed to help clarify the cytokine’s role in this process. In clinical studies, significantly elevated IFN-γ levels have been identified in both the serum and cerebrospinal fluid of CD19 CAR T-cell recipients during the acute phase of neurotoxicity, as compared to those in recipients without neurotoxicity [6,25,26,27,28].

### 3.3. Interleukin-1β

IL-1β is produced by activated monocytes and macrophages in response to CAR T cells recognizing CD19. IL-1β increases the BBB’s permeability by suppressing the production of SHH by astrocytes and, consequently, downregulating the expression of tight junction proteins [33]. In addition, IL-1β stimulates the release of vascular endothelial growth factor from astrocytes, which exacerbates the BBB disruption [34]. Once trafficked into the CNS, IL-1β stimulates glial cell proliferation by activating the mitogen-activated protein kinase pathway and induces further local cytokine production from both glial cells and microglia [35]. A key preclinical study by Norelli et al., in 2018, in a mouse model of ICANS treated with humanized CD19 CAR T cells, demonstrated that pre-treatment with IL-1β blockade prevented the development of signs of neurotoxicity, including generalized paralysis and seizures [36]. A clinical study of ICANS by Cohen et al., in 2019, reported elevated serum concentrations of IL-1β during the acute neurotoxicity phase, but none of the clinical studies evaluating cytokines in cerebrospinal fluid have identified IL-1β [6,25,26,27,28,37,38]. More clinical studies on the trends in serum and cerebrospinal fluid levels of IL-1β, before and during acute neurotoxicity, are needed to elucidate its role in the pathogenesis of ICANS.

### 3.4. Interleukin-6

IL-6 is produced by actively proliferating monocytes in CD19 CAR T–cell recipients and has been linked to the symptoms of cytokine release syndrome [19]. Preclinical studies by Saija et al. and Krizanac-Bengez et al. have demonstrated the increased permeability of the BBB in response to systemic IL-6 administration in rats [12,39]. Takeshita et al. investigated the mechanism of the altered permeability in human BBB models, noting a dose- and time-dependent decrease in the endothelial expression of tight junction proteins with IL-6 [40]. Despite this compelling evidence for the role of IL-6 in altering BBB permeability, preclinical models of ICANS have not yet established a direct causative role for IL-6 in ICANS. Norelli et al. demonstrated high serum levels of IL-6 during cytokine release syndrome and amelioration of the syndrome after treatment with the IL-6 blocker tocilizumab in mice engrafted with human leukemic blasts and treated with human CD19 CAR T cells. However, tocilizumab failed to protect mice from ICANS, which was ultimately fatal [36].

Several clinical studies of ICANS in both adult and pediatric patients have demonstrated elevated serum and cerebrospinal fluid levels of IL-6 during the acute neurotoxicity phase [6,25,26,27,28]. Once within the brain microenvironment, IL-6 plays a dual role: it suppresses TNF-α release and neutrophil migration early during inflammation, and it actively recruits monocytes and T cells during the late phase of inflammation. IL-6 also stimulates astrocytosis and angiogenesis, which are required for tissue remodeling during the late phase of inflammation [41].

### 3.5. Interleukin-10

The role of IL-10 in the pathophysiology of ICANS is somewhat controversial. Studies on inflammatory bowel disease have demonstrated the pro-inflammatory effect of IL-10, as evidenced by the activation of cytotoxic CD8+ T cells and subsequent tissue injury [42,43]. On the other hand, pre-clinical models have established the strong ant-inflammatory response of IL-10 by inhibiting cytokine production by both T-cells and natural killer cells [44]. In three clinical studies investigating the CSF levels of cytokines in patients with ICANS, elevated levels of IL-10 were noted in the post-infusion period, relative to the pre-infusion period. It is hypothesized that IL-10 may play a key role in maintaining the expansion and the activity of CAR T-cells. More detailed clinical studies comparing the CSF levels of IL-10 in patients with and without ICANS, as well as in those with mild vs. severe forms of ICANS may help to more definitively establish the role of IL-10 [6,26,28].

### 3.6. Interleukin-15

IL-15 plays a key role in the pathogenesis of cytokine release syndrome and ICANS in CD19 CAR T–cell recipients [19]. Preclinical studies on rat brain endothelium by Pan et al. in 2009 and Wu et al. in 2010 demonstrated an increased expression of IL-15 receptors on the surface endothelium in response to the systemic release of pro-inflammatory cytokines such as TNF-α. Increased receptor expression, in turn, leads to the endocytosis of IL-15 and trafficking across the BBB [45,46]. The role of IL-15 in neuroinflammation has been studied in preclinical models of multiple sclerosis. A preclinical study on human fetal neural tissue by Saikali et al. demonstrated an increased recruitment and activation of CD8 T lymphocytes in response to IL-15 and elevated levels of lytic enzymes and antigen-specific cytotoxicity, which play a key role in demyelination [47]. Preclinical models of ICANS which evaluate IL-15 are currently lacking, but the evidence of neuroinflammation in other models and the consistent finding of elevated serum IL-15 levels in patients with ICANS, which has been documented in clinical studies, strongly suggest that this cytokine has a role in in the etiology of ICANS. In addition, one clinical study demonstrated higher levels of IL-15 in cerebrospinal fluid from patients with ICANS, as compared to those at baseline [6]. More preclinical studies exploring the exact mechanism of IL-15 in neuroinflammation and more clinical studies on the changes in IL-15 levels in cerebrospinal fluid are needed to establish its role in the etiology of ICANS.

### 3.7. Angiopoeitin-2

Under conditions of vascular quiescence, angiopoeitin-1 is bound to the endothelial receptor tyrosine kinase Tie2 and maintains the integrity of the BBB. Angiopoeitin-2 is stored within endothelial Weibel–Palade bodies and is released during endothelial activation in response to pro-inflammatory cytokines. Angiopoietin-2 preferentially binds to Tie2 during periods of stress, thereby promoting platelet aggregation and thrombosis. In animal studies of cerebral malaria and traumatic brain injury, angiopoietin-2/Tie2 binding was associated with the destabilization of the BBB, whereas the manipulation of signaling towards the angiopoietin-1/Tie2 demonstrated a protective effect [48,49]. The role of angiopoietin-2 has also been investigated in CD19 CAR T–cell recipients who developed ICANS. In a study by Gust et al. in 2017, data from 133 adult CD19 CAR T–cell recipients demonstrated a positive correlation between the angiopoietin-2–angiopoietin-1 ratio and the grade of neurotoxicity [6]. In addition, this study demonstrated that patients with grade 4 or higher neurotoxicity have higher serum levels of von Willebrand factor than those with grade 3 or lower neurotoxicity. The von Willebrand factor is stored in the Weibel–Palade bodies of the endothelium with angiopoeitin-2 and is a key stimulator of platelet aggregation [6]. These findings suggest the role of the angiopoietin-2/Tie2 axis in BBB destabilization; however, more prospective clinical studies are needed to establish the role of antiopoeitin-2 in the pathogenesis of ICANS.

## 4. Clinical and Imaging Features of ICANS

The CAR T–cell- and cytokine-mediated BBB breakdown and the subsequent neuroinflammation can be associated with various clinical features, ranging from mild encephalopathy to seizures and cerebral edema. The development of ICANS is monophasic, with symptoms appearing within 3–6 days post CAR T-cell infusion, peaking at days 7–8 and resolving by days 14–21 [50]. The most frequently reported clinical manifestation of ICANS is cognitive dysfunction, which has been variably described in the literature as delirium, confusion, or encephalopathy [7,26,27]. The cognitive dysfunction in ICANS is often marked by a striking disturbance in speech and handwriting. Although headaches and tremors are non-specific symptoms, they frequently precede or accompany the classic cognitive dysfunction in ICANS [8]. The incidence of seizures in ICANS ranges widely from 5–10%, with a slightly higher incidence rate in recipients of CAR T-cell products with the CD28 co-stimulatory domain, as compared to the 4-1BB co-stimulatory domain [51]. Fatal cerebral edema occurs in 1–2% of CD19 CAR T-cell recipients and is notably absent in recipients of other antigen-specific CAR T-cell products such as CD22. Many of the mechanisms of neurotoxicity, such as the microvascular disruption and endothelial activation described earlier, have been described in cases of fatal cerebral edema and it is unclear whether milder forms of ICANS represent a less profound version of the same mechanism, or a completely different mechanism. More research is necessary to help us to clearly identify the pathophysiologic mechanisms of the varying grades of ICANS [52]. The current grading of ICANS incorporates cognition, consciousness, motor function, seizure activity, and signs of elevated intracranial pressure (Table 1) [8]. Several clinical studies in both adult and pediatric CD19 CAR T-cell recipients have described elevated serum cytokine levels in patients with ICANS, but studies comparing the CSF levels of these cytokines in patients with and without ICANS are currently lacking due to a lack of a clinical indication for obtaining CSF in patients without ICANS [6,26,27]. While obtaining CSF in all CD19 CAR T-cell recipients would certainly help characterize the temporal evolution of inflammation in the CNS better, the practical considerations are currently a limiting factor.

Despite the specific and temporal nature of clinical symptoms, conventional MR imaging of the brain is often non-contributory. Most patients with grade 1–2 ICANS have no acute imaging abnormalities in the first 21 days post-infusion, and a few of the patients with grade 3–4 ICANS demonstrate non-specific imaging abnormalities. However, these abnormalities may provide a valuable insight into the underlying pathophysiological mechanisms of ICANS, and radiologists need to be aware of their implications [53]. A small number of patients may present with diffuse leptomeningeal enhancement, a finding which may be seen in infectious or chemical meningitis. In the setting of recent CAR T-cell infusion, however, the findings are likely reflective of a breakdown of the BBB, secondary to the targeting of CD19 on the pericytes and the cytokine-mediated endothelial activation [54]. Another neuroimaging finding reported in ICANS is non-specific hyperintensities on the T2 and fluid attenuated inversion recovery (FLAIR) sequences in the cerebral and cerebellar white matter, which may be linked to the demyelinating effects of IFN-γ. The cytokine induced neuronal excitotoxicity may result in transient restricted diffusion in the cerebral cortex or the splenium of the corpus callosum, due to a higher concentration of excitotoxic neurons in these locations. These findings must be interpreted in the context of recent CAR T-cell infusion and new neurologic symptoms. Other findings reported on MRIs include hyperintensity on the T2 and/or FLAIR sequences in the thalami, brainstem and the external capsules, likely reflecting a combination of local microvascular disruption and excitotoxicity [55]. More research using advanced imaging modalities, such as dynamic susceptibility weighted perfusion imaging to quantify local changes in perfusion and diffusion tensor imaging to quantify changes in white matter anisotropy, are needed to support these hypotheses and establish more definitive imaging criteria for ICANS.

In the 1–2% of CD19 CAR T-cell recipients with fatal cerebral edema, MRI is often not feasible due to clinical instability. A non-contrast head CT is often the only imaging modality that can be performed safely and in a timely manner, and the interpreting radiologist plays a critical role in the management of these patients. Recognizing the early signs of cerebral edema on CT can often alert the clinical team to the possibility of this dreaded complication and implement early therapy with steroids and interleukin antagonists. The early radiographic signs are subtle and non-specific, with mild periventricular white matter hypoattenuation and sulcal effacement suggesting edema development. In the severe form of cerebral edema, there is marked, diffuse cerebral hypoattenuation with the loss of gray-white matter differentiation, effacement of the ventricles, sulci and cisterns, and even frank herniation [55].

## 5. Conclusions

The BBB is a complex neurovascular unit, with a unique susceptibility to CD19 CAR T–cell therapy at the cellular level. Knowledge of the off-target mechanism of CD19 CAR T cells on pericytes and the role of individual cytokines in decreasing the expression of endothelial tight junction proteins, increasing the expression of cellular adhesion molecules, and trafficking activated monocytes across the BBB has helped further our understanding of the complex neurotoxicity that sometimes accompanies this therapy. Preclinical models of ICANS are particularly needed to help us better understand the role of certain cytokines (e.g., IL-15 and angiopoeitin-2) in BBB breakdown and subsequent neuroinflammation. Furthermore, more prospective clinical studies that investigate the trends in cytokine levels in cerebrospinal fluid are needed to help us understand their temporal relations with the onset and duration of ICANS.

## Figures and Tables

**Figure 1 ijms-25-00644-f001:**
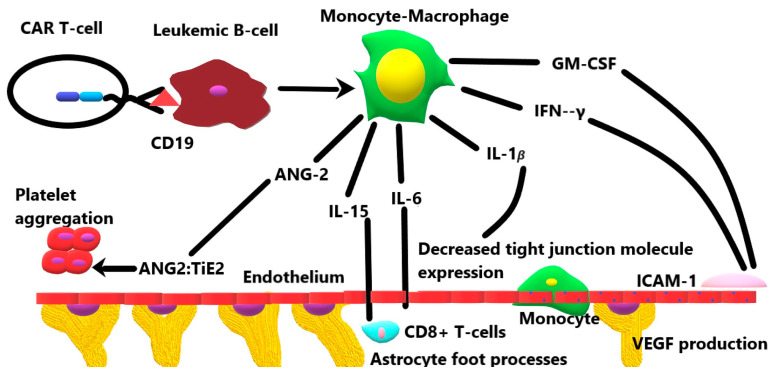
The role of various cytokines released by the activation of the monocyte–macrophage system in response to CD19 CAR T-cell therapy.

**Table 1 ijms-25-00644-t001:** The 2018 American Society of Transplantation and Cellular Therapy criteria for grading of ICANS [8].

Neurotoxicity Domain	Immune Effector Cell–Associated Neurotoxicity Syndrome Grades			
	1	2	3	4
ICE * score	7–9	3–6	0–2	0
CAPD ^†^ score	1–8	1–8	≥9	Unable to assess
Level of consciousness	Awakens spontaneously	Awakens to voice	Awakens to tactile stimulus	Stupor or coma
Motor weakness	None	None	None	Hemiparesis/paraparesis
Seizure	None	None	Any seizure that resolves rapidly or nonconvulsive seizure that resolves with intervention	Prolonged or life-threatening seizure lasting > 5 min, or repetitive seizures without return to baseline
Intracranial hypertension/cerebral edema	None	None	Focal/local edema on neuroimaging	Decerebrate/decorticate posturing, cranial nerve VI palsy, papilledema, Cushing’s triad, diffuse cerebral edema on imaging.

* The Immune Effector Cell–Associated Encephalopathy (ICE) score is a cognitive scale used to assess children aged 12 years or older. Scores range from 0 (unarousable) to 10 (no impairment) and incorporate orientation, attention, ability to name, follow commands, and write. ^†^ The Cornell Assessment for Pediatric Delirium (CAPD) score is a cognitive scale used to assess children younger than 12 years old. Scores range from 0–8 and incorporate components of the child’s interaction with his/her surroundings. Scores of 9 or more indicate a poor prognosis.

## Data Availability

No new data were created or analyzed in this study. Data sharing is not applicable to this article.

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
