# Peer review of "The Mechanisms of Altered Blood–Brain Barrier Permeability in CD19 CAR T–Cell Recipients"

_ijms, 2024, doi:10.3390/ijms25010644_

Round 1

Reviewer 1 Report

Comments and Suggestions for Authors

The current review entitled “The mechanisms of altered blood–brain barrier permeability in 2 CD19 CAR T–cell recipients” summarized the physiology of BBB and how its permeability is affected by CD19 CAR T cells which are used as immunotherapy for B-cell malignancies. Although the precise reason for the changed BBB permeability during CD19 CAR T-cell treatment is unknown, several mechanisms are believed to be involved, including cytokine release and CAR-T direct neurotoxicity.

Pinto and Krenciute highlighted the underlying mechanisms and cytokines involved in the pathogenesis of immune effector cell–associated neurotoxicity syndrome (ICANS). This condition is a serious neurological complication of CD19 CAR T-cell therapy. The authors emphasized the role of each cytokine; GM-CSF, IFN‐γ, IL-1β, IL-6, IL-10, IL-15, and angiopoietin-2 in ICAM and they linked these changes with clinical markers and radiological features of ICANS. The research gaps are also mentioned throughout the manuscript. In my opinion, the authors selected a very interesting topic and reviewed previous work in reasonable depth and used relevant references with limited self-citation.

My only request is to draw a figure that illustrates the role of inflammatory cytokines in ICANS.

Author Response

Thank you for your kind words and suggestion to upload a figure.

I have included a figure highlighting the role of the various interleukins in affecting the permeability of the blood-brain barrier see figure 1.

Reviewer 2 Report

Comments and Suggestions for Authors

Dear Authors,

The review article provides a clear and concise overview of the topic, highlighting the significance of Cluster of Differentiation 19 (CD19) chimeric antigen receptor (CAR) T cells as a promising immunotherapy for B-cell malignancies. It effectively introduces the main challenge associated with this therapy, which is immune effector cell-associated neurotoxicity syndrome (ICANS). I have few questions to add.

1. Provide a bit more context on why CD19 CAR T-cell therapy is used and what conditions it aims to treat. This could help readers who are not experts in the field understand the significance of the discussion.

2. Please provide a cartoon model for a better understanding of the concept.

3. Elaborate on why prospective clinical studies on cytokine levels in cerebrospinal fluid are essential. What specific insights or clinical applications could arise from this research?

4. It would be helpful to mention if the review includes any visual aids or analyses that can assist in overcoming this diagnostic challenge.

Comments on the Quality of English Language

Minor correctors are required for the English corrections

Author Response

Thank you for your kind words and valuable suggestions.

  1. I have added a sentence in yellow highlight, specifying the populations that are treated with, and most likely to benefit from CD19 CAR T-cells in the introduction..
  2. I have added a cartoon model (Figure 1), that briefly summarizes the roles of the various cytokines in ICANS.
  3. I have added a sentence in the clinical and imaging section, in yellow highlight suggesting the challenges of obtaining CSF in the non ICANS population.
  4. I am not sure what you mean by visual aids, I think figure 1 is helpful.